# A community approach to mortality prediction in sepsis via gene expression analysis

Timothy E. Sweeney [1,2,25], Thanneer M. Perumal [3], Ricardo Henao [4,5], Marshall Nichols[4], Judith A. Howrylak[6], Augustine M. Choi[7], Jesús F. Bermejo-Martin[8], Raquel Almansa[8], Eduardo Tamayo[8], Emma E. Davenport [9,10,11], Katie L. Burnham[12], Charles J. Hinds [13], Julian C. Knight[12], Christopher W. Woods[4,14,15], Stephen F. Kingsmore[16], Geoffrey S. Ginsburg[4], Hector R. Wong[17,18], Grant P. Parnell[19], Benjamin Tang[19,20,21,22], Lyle L. Moldawer[23], Frederick E. Moore[23], Larsson Omberg[3], Purvesh Khatri [1,2], Ephraim L. Tsalik [4,14,15], Lara M. Mangravite [3] & Raymond J. Langley [24]

Improved risk stratification and prognosis prediction in sepsis is a critical unmet need. Clinical severity scores and available assays such as blood lactate reflect global illness severity with suboptimal performance, and do not specifically reveal the underlying dysregulation of sepsis. Here, we present prognostic models for 30-day mortality generated independently by three scientific groups by using 12 discovery cohorts containing transcriptomic data collected from primarily community-onset sepsis patients. Predictive performance is validated in five cohorts of community-onset sepsis patients in which the models show summary AUROCs ranging from 0.765–0.89. Similar performance is observed in four cohorts of hospital-acquired sepsis. Combining the new gene-expression-based prognostic models with prior clinical severity scores leads to significant improvement in prediction of 30-day mortality as measured via AUROC and net reclassification improvement index These models provide an opportunity to develop molecular bedside tests that may improve risk stratification and mortality prediction in patients with sepsis.

[1] Stanford Institute for Immunity, Transplantation and Infection, Stanford University School of Medicine, Stanford, CA 94305, USA. [2] Division of Biomedical Informatics Research, Department of Medicine, Stanford University School of Medicine, Stanford, CA 94305, USA. [3] Sage Bionetworks, Seattle, WA 98109, USA. [4] Center for Applied Genomics and Precision Medicine, Department of Medicine, Duke University, Durham, NC 27708, USA. [5] Department of Electrical and Computer Engineering, Duke University, Durham, NC 27708, USA. [6] Division of Pulmonary and Critical Care Medicine, Penn State Milton S. Hershey Medical Center, Hershey, PA 17033, USA. [7] Department of Medicine, Cornell Medical Center, New York, NY 10065, USA. [8] Hospital Clínico Universitario de Valladolid/IECSCYL, Valladolid 47005, Spain. [9] Department of Medicine, Brigham and Women's Hospital, Harvard Medical School, Boston, MA 02115, USA. [10] Partners Center for Personalized Genetic Medicine, Boston, MA 02115, USA. [11] Program in Medical and Population Genetics, Broad Institute of MIT and Harvard, Cambridge, MA 02142 USA. [12] Wellcome Trust Centre for Human Genetics, University of Oxford, Oxford OX3 7BN, UK. [13] William Harvey Research Institute, Barts and The London School of Medicine, Queen Mary University, London EC1M 6BQ, UK. [14] Division of Infectious Diseases and International Health, Department of Medicine, Duke University, Durham, NC 27710, USA. [15] Durham Veteran's Affairs Health Care System, Durham, NC 27705, USA. [16] Rady Children's Institute for Genomic Medicine, San Diego, CA 92123, USA. [17] Division of Critical Care Medicine, Cincinnati Children's Hospital Medical Center and Cincinnati Children's Research Foundation, Cincinnati, OH 45223, USA. [18] Department of Pediatrics, University of Cincinnati College of Medicine, Cincinnati, OH 45267, USA. [19] Centre for Immunology and Allergy Research, Westmead Institute for Medical Research, Westmead, NSW 2145, Australia. [20] Department of Intensive Care Medicine, Nepean Hospital, Sydney, Australia, Penrith, NSW 2751, Australia. [21] Nepean Genomic Research Group, Nepean Clinical School, University of Sydney, Penrith, NSW 2751, Australia. [22] Marie Bashir Institute for Infectious Diseases and Biosecurity, Westmead NSW 2145, Australia. [23] Department of Surgery, University of Florida College of Medicine, Gainesville, FL 32610, USA. [24] Department of Pharmacology, University of South Alabama, Mobile, AL 36688, USA. [25] Inflammatix Inc., Burlingame, CA 94010, USA. Timothy E. Sweeney and Thanneer M. Perumal contributed equally to this work. Purvesh Khatri, Ephraim L. Tsalik, Lara M. Mangravite and Raymond J. Langley jointly supervised this work. Correspondence and requests for materials should be addressed to R.J.L. (email: rlangley@southalabama.edu)

S epsis, recently defined as organ dysfunction caused by a dysregulated host response to infection[1], contributes to half of all in-hospital deaths in the US and is the leading cost for the US healthcare system[2,3]. Although in-hospital sepsis outcomes have improved over the last decade with standardized sepsis care, mortality rates remain high (10–35%)[4]. Sepsis treatment still focuses on general management strategies including source control, antibiotics, and supportive care. Despite dozens of clinical trials, no treatment specific for sepsis has been successfully utilized in clinical practice[5]. Two consensus papers suggest that continued failure of proposed sepsis therapies is due to substantial patient heterogeneity in the sepsis syndrome and a lack of tools to accurately categorize sepsis at the molecular level[5,6]. Current tools for risk stratification include clinical severity scores such as APACHE or SOFA as well as blood lactate levels. While these measures assess overall illness severity, they do not adequately quantify the patient's dysregulated response to the infection and therefore fail to achieve the personalization necessary to improve sepsis care[7]. Some peptide markers of sepsis severity have been validated (e.g. proadrenomedullin[8] among others[9]), but these are not yet cleared for clinical use.

A molecular definition of the severity of the host response in sepsis would provide several benefits. First, improved accuracy in sepsis prognosis would improve clinical care through appropriate matching of patients with resources: the very sick can be diverted to intensive care unit (ICU) for maximal intervention, while patients predicted to have a better outcome may be safely watched in the hospital ward or discharged early. Second, more-precise estimates of prognosis would allow for better discussions regarding patient preferences and the utility of aggressive interventions. Third, better molecular phenotyping of sepsis patients has the potential to improve clinical trials through both (1) patient selection and prognostic enrichment for drugs and interventions (e.g., excluding patients predicted to have good vs. bad outcomes) and (2) better assessments of observed-to-expected ratios for mortality[5,6]. Finally, as a direct quantitative measure of the dysregulation of the host response, molecular biomarkers could potentially help form a quantitative diagnosis of sepsis as distinct from non-septic acute infections[10,11]. Thus, overall, a quantitative test for sepsis could be a significant asset to clinicians if deployed as a rapid assay.

Previously, a number of studies have used whole-blood transcriptomic (genome-wide expression) profiling to risk-stratify sepsis patients[12–15]. Important insights from these studies suggest that more-severe sepsis is accompanied by an overexpression of neutrophil proteases, adaptive immune exhaustion, and an overall profound immune dysregulation[12,13,16–19]. Quantitative evaluation of host response profiles based on these observations has been validated prospectively to show specific outcomes[14,15], but none have yet been translated into clinical practice. Still, the availability of high-dimensional transcriptomic data from these accumulated studies has created unprecedented opportunities to address questions across heterogeneous representations of sepsis (different ages, pathogens, and patient types) that could not be answered by any individual cohort.

Transcription-based modeling has been deployed across many diseases to improve prognostic accuracy. These are typically developed in a method-specific manner using data collected from single cohorts. As a result, prognostic models often lack the generalizability that is necessary to confer utility in clinical applications[20]. In contrast, community modeling approaches (where multiple groups create models using the same training data) can provide an opportunity to explicitly evaluate predictive performance across a diverse collection of prognostic models sampled from across a broad solution space[21–25]. Here, we systematically identified a large collection of both public and privately held gene expression data from clinical sepsis studies at the time of sepsis diagnosis. Three scientific groups were then invited to build models to predict 30-day mortality based on gene expression profiles. These three groups produced four different prognostic models, which were then validated in external validation cohorts composed of patients with either community-acquired sepsis or hospital-acquired infections (HAIs).

## Results

**Analysis overview.** We used a community approach to build gene-expression-based models predictive of sepsis-induced mortality using all available gene expression datasets (21 total cohorts, Table 1). In this community approach, three different teams (Duke University, Sage Bionetworks, and Stanford University) performed separate analyses using the same input data; we thus sampled the possible model space to determine whether output performance is a function of analytical approaches (Fig. 1). Two models (Duke and Stanford) used parameter-free difference-of-means formula for integrating gene expression, and the other two models (both from Sage Bionetworks) used parametrized penalized logistic regression (LR)[26] and random forests (RF)[27].

Each of the four models was trained using 12 discovery cohorts (485 survivors and 157 non-survivors) composed primarily of patients with community-acquired sepsis. Performance was evaluated across two groups of heterogeneous validation datasets (five community-acquired sepsis cohorts with 161 survivors and 28 non-survivors and 4 HAI cohorts with 258 survivors and 24 non-survivors, Table 1). The community-acquired sepsis and HAI cohorts were considered separately in validation because of their known differences in host-response profiles. Due to the nature of public datasets, we had limited information on demographics, infection, severity, and treatment and so these variables were not controlled for in model selection. The cohorts included patients from multiple age groups, countries, and hospital wards (emergency department, hospital ward, medical ICU, and surgical/trauma ICU). As expected in varied patient populations, mortality rates varied widely across cohorts (mean 23.2% ± 13.4%).

**Prognostic power assessments.** Model performance was primarily evaluated using receiver operating characteristic (ROC) analysis separately in the discovery, validation, and HAI cohorts. Boxplots of the AUROCs for each model are shown in Fig. 2; data from individual cohorts and summary ROC curves are shown in Supplementary Tables 1 and 2 and Supplementary Fig. 4. Across the five community-acquired sepsis validation datasets, the four models showed generally preserved prognostic power, with summary AUROCs ranging from 0.75 (95% CI 0.63–0.84, Sage LR) to 0.89 (95% CI 0.56–0.99, Stanford). Three of the four models performed well in classifying the HAI datasets (summary AUROCs 0.81–0.87 in the Duke, Sage LR, and Stanford models); one model performed poorly in HAI (summary AUROC 0.52, 95% CI 0.36–0.68, Sage RF). Overall, most models performed equivalently in discovery, validation, and HAI datasets. To judge other performance metrics including accuracy, specificity, negative predictive value, and positive predictive value, we set thresholds for each model at the nearest sensitivity >90% (Supplementary Fig. 5). The raw prediction scores for each sample in each model are available for further interpretation[28].

Using the validation and HAI cohorts, we compared the present models to a single prognostic model made with all genes previously associated with mortality (see Supplementary Methods)[13,17–19,29,30]. We found that that three of the four models show substantial improvement (average increase of

## Table 1 Datasets included in the analysis

| Dataset accession | First author | Cohort description | Timing of sepsis diagnosis | Percent bacterial infection | Age | Sex (% male) | Severity | Country | No. survived | No. died |
|---|---|---|---|---|---|---|---|---|---|---|
| **1a: Discovery Cohorts** | | | | | | | | | | |
| E-MEXP-3567 | Irwin | Children with meningococcal sepsis +/− HIV co-infection | Admission to ED | 100 | 2.0 (IQR 0.6–6.9) | 55 | unk. | Malawi | 6 | 6 |
| E-MEXP-3850 | Kwan | Children w/ meningococcal sepsis | Admission to hospital; sampled at multiple times 0–48 h | 100 | 1.3 (range 0.8–2.0) | 40 | PELOD; 29.2 (range 11–61) | UK | 19 | 5 |
| E-MTAB-1548 | Almansa | Adult surgical patients with sepsis (EXPRESS study) | Average post-operation day 4 (hospital acquired) | 100 | 69.7 (std. dev. 13.1) | 67 | APACHE II 17.0 (std. dev. 5.4) | Spain | 50 | 24 |
| GSE10474 | Howrylak | Adults in MICU with sepsis +/− ALI | Admission to ICU | 75+ | 57 (std. dev. 4.3) | 45 | APACHE II 20.7 (std. dev. 1.6) | USA | 22 | 11 |
| GSE13015a GSE13015b | Pankla | Adults with sepsis, many from burkholderia | Within 48 h of diagnosis; both community-acquired and hospital-acquired | 100 | 54.7 (std. dev. 11.7) | 54 | unk. | Thailand | 35 8 | 13 7 |
| GSE27131 | Berdal | Adults with severe H1N1 influenza requiring mechanical ventilation | Admission to ICU | 0 | unk. | unk. | SAPS II 29.3 (std. dev. 10.3) | Norway | 5 | 2 |
| GSE32707 | Dolinay | Adults in MICU with sepsis+/− ARDS | Admission to ICU | unk. | 57.1 (std. dev. 14.9) | 53 | APACHE II 26.7 (std. dev. 8.5) | USA | 31 | 17 |
| GSE40586 | Lill | Infants, children, and adults with bacterial meningitis | Within 48 h of hospital admission | 100 | 43.4 (range 17 days –70 years) | unk. | unk. | Estonia | 19 | 2 |
| GSE63042 | Langley | Adults with sepsis (CAPSOD study) | Admission to ED | 80+ | 59.1 (std. dev. 18.3) | 59 | APACHE II 16.5 (std. dev. 7.3) | USA | 76 | 28 |
| GSE66099 | Wong | Children in ICU with sepsis/septic shock | Admission to ICU | 72 | 3.7 | 58 | PRISM 15.7 | USA | 171 | 28 |
| GSE66890 | Kangelaris | Adults in ICU with sepsis +/− ARDS | Admission to ICU | | 63 (std. dev 19) | 56 | APACHE III 100 (std. dev. 35) | USA | 43 | 14 |
| **1b: Validation cohorts** | | | | | | | | | | |
| GSE21802 | Bermejo-Martin | Adults in ICU with severe H1N1 influenza | Within 48 h of admission to ICU | 0 | 43 (std. dev. 11) | 47 | SOFA 4.1 (std. dev. 3.5) | Spain | 7 | 4 |
| GSE33341 | Ahn | Adults with 2+ SIRS criteria and bacteremia | Within 24 h of admission to hospital | 100 | 58 (range 24–91) | 61 | unk. | USA | 49 | 2 |
| GSE54514 | Parnell | Adults in ICU with sepsis | Admission to ICU | unk. | 61 (std. dev. 16) | 40 | APACHE II 21 (std. dev. 6) | Australia | 26 | 9 |
| GSE63990 | Tsalik | Adults with bacterial infection plus 2+ SIRS criteria | Admission to ED | 100 | 49 (range 14–88) | 50 | unk. | USA | 64 | 6 |
| E-MTAB-4421.51 | Davenport | Adults with sepsis (GAinS study) | Day of hospital admission | 92 | 64.2 (std. dev. 15.2) | 55 | APACHE II 18.6 (std. dev. 9.7) | UK | 15 | 7 |
| **1c: Hospital-acquired infection cohorts** | | | | | | | | | | |
| Duke HAI | Tsalik (unpublished) | Adults who developed ventilator-associated pneumonia (VAP) | Hospital days 1–30 | unk. | 58.0 (std. dev. 17.9) | 75 | unk. | USA | 60 | 10 |
| Glue Grant Burns | Glue Grant authors | Adults with severe burns (whole blood) | Hospital days 1–30 | 100 | 14.1 (std. dev. 16.2) | 64 | Denver Score 1.5 (S 1.7) | USA | 84 | 8 |
| Glue Grant Trauma | Glue Grant authors | Adults with severe traumatic injuries (buffy coat) | Hospital days 1–30 | 100 | 33.2 (std. dev. 10.2) | 74 | MODS 6.4 (std. dev. 3.3) | USA | 48 | 1 |
| UF P50 12H | Moldawer (unpublished) | Adults with hospital-acquired sepsis | Hospital days 1–30 | 100 | unk. | unk. | SOFA 5.5 (std. dev. 3.9) | USA | 66 | 5 |

Unk, unknown data or not available; IQR, inter-quartile range; std. dev., standard deviation; ED, emergency department; ICU, intensive care unit; MICU, medical ICU; ARDS, acute respiratory distress syndrome; SIRS, systemic inflammatory response syndrome; VAP, ventilator-associated pneumonia

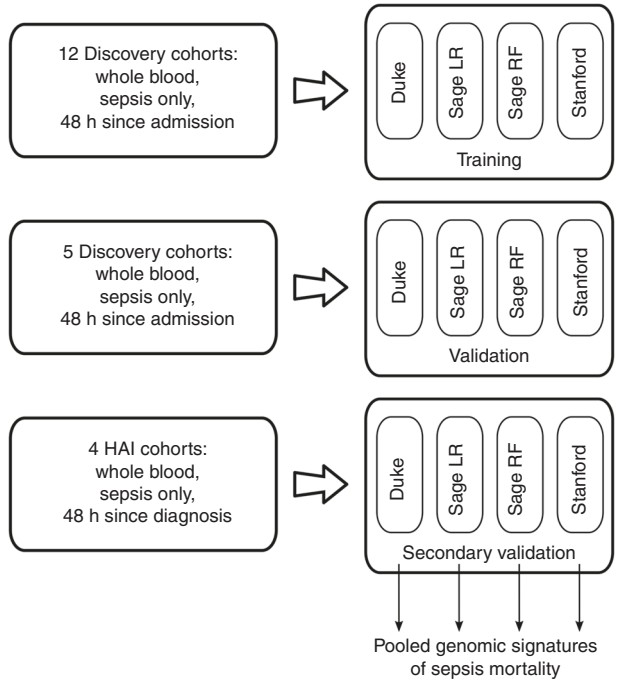

**Fig. 1** Overview of analysis: schema of our community-modeling-based approach to multi-cohort analysis. Three phases are shown, as described in the Methods section: (i) discovery, (ii) validation, and (iii) secondary validation (HAI cohorts)

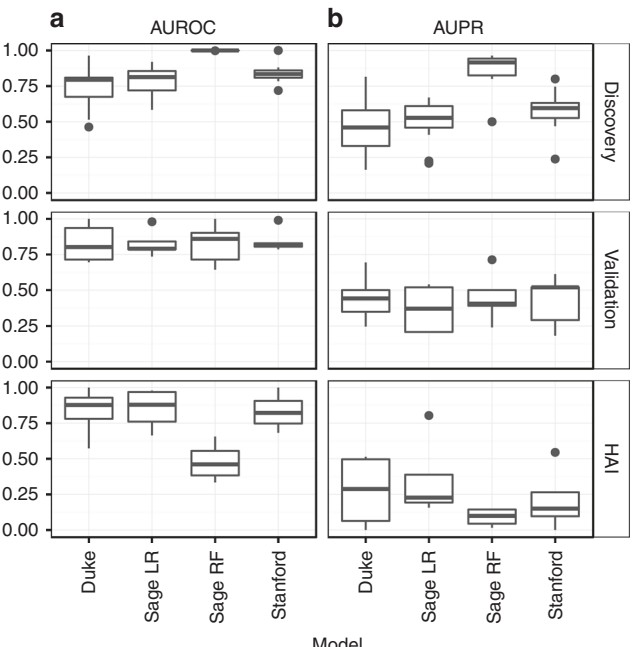

**Fig. 2** Model performance of the four genomic mortality predictors as measured by (**a**) AUROC and (**b**) AUPRC. The three panels (top, middle, bottom) show boxplots of the performance across all Discovery, Validation, and HAI cohorts

roughly 0.1) compared to the prior models; this reached significance for the Duke and Stanford models (Supplementary Table 3).

To assess whether the models contained complementary orthogonal information, we evaluated the prediction accuracy of an ensemble model based on the predictions of all four individual models (see Supplementary Methods). The prognostic power of the ensemble model was at an average AUROC of 0.81 across all five validation datasets (paired $t$-tests vs. individual models all $P =$ NS, Supplementary Table 4) indicating that the present diagnostic accuracy may be a rough estimate of the ceiling of prognostic accuracy inherent in these data.

Performance in predicting non-survivors was evaluated using the area under the precision–recall curve (AUPRC)[31] (Fig. 2b and Supplementary Table 5). The AUPRCs for non-survivor prediction were notably lower than the AUROCs, as was expected from the highly unbalanced classes (rare mortalities). This suggests that the models' primary utility may be in ruling out mortality for individuals much less likely to die within 30 days (those less likely to require substantial intervention) as opposed to accurately identifying the minority of patients who are highly likely to die within 30 days. On the contrary the AUPRC of the ensemble model was averaged at 0.428 in validation cohorts (Supplementary Table 4), suggesting complementarity in discriminatory power between individual models.

We examined the effects of clinical time course on the gene scores in the two validation datasets that tracked longitudinal data (GSE21802 and GSE54154; Supplementary Fig. 6). We found no differences in slope (change in score over time) between the survivors and non-survivors, although the scores in non-survivors were significantly higher than in survivors during the entire hospital stay, possibly indicating a failure to restore homeostasis.

**Comparison to standard predictors**. We next assessed whether the performance of these gene expression-based predictors of

mortality outperformed standard clinical severity scores. Notably, clinical measures of severity were only available in a subset of cohorts (eight discovery, three validation, three HAI; Table 2). The mean differences in the AUROCs of the gene model over clinical severity scores were: Duke −0.044; Sage LR 0.010; Sage RF 0.094; Stanford 0.064; only the Stanford model trended towards significance (paired $t$-test $P = 0.098$). However, we combined gene models and clinical severity scores into joint predictors, and each combination significantly outperformed clinical severity scores alone (mean difference Duke 0.077; Sage LR 0.076; Sage RF 0.16; Stanford 0.098; all paired $t$-tests $p \leq 0.01$).

We next examined continuous net reclassification improvement (cNRI) index to quantify how well the model with gene scores reclassifies survivors over the model with clinical severity scores in each of these same datasets (Table 3). In the validation and HAI cohorts, the mean NRI was 0.53–0.84 (potential range 0–2, where 2 reflects all patients with improved classification). For the Duke and Stanford scores, half of the validation and HAI datasets showed significant NRI compared to standard predictors alone. This suggests that the gene expression-based predictors add significant prognostic utility to standard clinical metrics.

Finally, we examined test characteristics at a high-sensitivity cutoff (95%) and a high-specificity cutoff (95%) for the gene scores in comparison to baseline error models (Supplementary Table 6) and in comparison to clinical severity scores (Supplementary Table 7). Overall mean accuracy of the joint clinical and gene scores was higher in the validation and HAI datasets (0.58–0.72 and 0.64–0.79 across the models, respectively) compared to clinical scores alone (0.57 and 0.62, respectively).

**Comparison across models**. We next studied whether models were correctly classifying the same patients or different groups of patients. We tested model correlations across all patients by comparing the relative ranks of each patient within each model instead of comparing raw model scores. We found the models were moderately correlated (Spearman rho = 0.35 − 0.61,

**Table 2 AUROC with genomic features and clinical severity**

| Dataset | Score type | Severity alone | Duke | | Sage LR | | Sage RF | | Stanford | |
|---|---|---|---|---|---|---|---|---|---|---|
| | | | Gene model alone | Joint model | Gene model alone | Joint model | Gene model alone | Joint model | Gene model alone | Joint model |
| **Discovery datasets** | | | | | | | | | | |
| EMEXP3850 | PELOD | 1 | 0.947 | 1 | 0.916 | 1 | 1 | 1 | 1 | 1 |
| EMTAB1548 | SOFA | 0.735 | 0.817 | 0.843 | 0.863 | 0.87 | 1 | 1 | 0.849 | 0.863 |
| GSE10474 | APACHE II | 0.551 | 0.53 | 0.626 | 0.682 | 0.758 | 1 | 1 | 0.722 | 0.697 |
| GSE27131 | SAPS II | 1 | 0.7 | 1 | 0.7 | 1 | 1 | 1 | 1 | 1 |
| GSE32707 | APACHE II | 0.546 | 0.514 | 0.537 | 0.712 | 0.702 | 0.996 | 0.996 | 0.81 | 0.805 |
| GSE63042 | APACHE II | 0.774 | 0.679 | 0.797 | 0.866 | 0.868 | 1 | 1 | 0.742 | 0.815 |
| GSE66099 | PRISM | 0.781 | 0.806 | 0.84 | 0.916 | 0.913 | 1 | 1 | 0.881 | 0.892 |
| GSE66890 | APACHE II | 0.723 | 0.802 | 0.847 | 0.711 | 0.759 | 1 | 1 | 0.834 | 0.849 |
| **Validation datasets** | | | | | | | | | | |
| EMTAB4421 | APACHE | 0.705 | 0.695 | 0.771 | 0.81 | 0.762 | 0.714 | 0.752 | 0.829 | 0.838 |
| GSE21802 | SOFA | 0.812 | 0.333 | 0.833 | 0.708 | 0.792 | 0.583 | 0.833 | 0.75 | 0.833 |
| GSE54514 | APACHE | 0.776 | 0.936 | 0.944 | 0.701 | 0.739 | 0.902 | 0.927 | 0.816 | 0.825 |
| **HAI datasets** | | | | | | | | | | |
| Glue Burns D1-D30 | Denver score | 0.482 | 0.808 | 0.842 | 0.721 | 0.731 | 0.606 | 0.604 | 0.74 | 0.756 |
| Glue Trauma D1-D30 | MODS score | 0.927 | 1 | 1 | 0.938 | 0.979 | 0.667 | 0.958 | 1 | 1 |
| UF P50 12H | SOFA | 0.941 | 0.573 | 0.945 | 0.652 | 0.945 | 0.6 | 0.952 | 0.682 | 0.945 |

Some gene model AUCs may differ from Supplementary Table 2 since samples without severity scores were dropped from this analysis

Supplementary Fig. 7). We then evaluated the agreement between the four models by comparing model-specific patient classifications (Supplementary Table 8). For this purpose, we chose cutoffs for each model that yielded 90% sensitivities for non-survivors. We then labeled patients as being either always misclassified, correctly classified by 1 or 2 models (no consensus), or correctly classified in at least 3 of 4 models (consensus). As expected by the 90% sensitivity threshold, 10% of patients were misclassified by all models. In the remaining cases, 63% were correctly predicted by consensus and 27% do not reach consensus. Together, the model correlation and consensus analyses showed that 73% of patients were classified by at least one model, with variance leading to discordance in the remaining 27%. These results suggest that although the models use different genes, they are reaching the same conclusions about most patients.

**Biology of the gene signatures of mortality**. Gene predictors were chosen for both optimized prognostic power and sparsity in our data-driven approach and so do not necessarily represent key nodes in the pathophysiology of sepsis. Still, we examined whether interesting biology was being represented in the models. We first looked for overlap in the gene sets used for prediction across the four models, but found few genes in common (Table 4). Since each signature had too few genes for robust analysis, we analyzed the genes from all four models in aggregate, resulting in 58 total genes (31 upregulated and 27 downregulated; Supplementary Table 9).

First, we studied whether the differential gene expression identified may be indicative of cell-type shifts in the blood. The pooled gene sets were tested in several known in vitro gene expression profiles of sorted cell types to assess whether gene expression changes are due to cell-type enrichment (Supplementary Fig. 8). No significant differences were found, but the trend showed an enrichment of M1-polarized macrophages and band cells (immature neutrophils), and underexpression in dendritic cells. This is consistent with a heightened pro-inflammatory response and a decrease in adaptive immunity in patients who ultimately progress to mortality[12].

We next tested the 58 genes for enrichment in curated gene sets from gene ontologies, Reactome and KEGG pathways using two different enrichment methodologies: gene-based over-representation analysis and expression-based GSEA. After multiple hypothesis testing corrections, 4 out of 3330 gene sets tested were significantly over-represented at an FDR of 5% (Supplementary Table 10a). These include genes related to the regulation of T cell activation and proliferation, cytokine-mediated signaling pathway and RHO GTPases activation of CIT. The relatively low number of pathways enriched in over-representation analysis may be due to the low number of genes in the predictor set. Enrichment of 58 gene predictors' expression were also tested using GSEA. Out of 1576 curated pathways, 546 were enriched at an FDR of 5%; top pathways are shown in Supplementary Table 10b. A brief examination of enriched pathways activated in non-survivors showed mostly inflammation-related pathways, while survivors showed largely developmental pathways. Since the models were generated in a way that penalized the inclusion of genes that were redundant for classification purposes, and since genes redundant for classification purposes are often from the same biological pathway, their exclusion from the models limits the utility of enrichment analyses.

## Discussion

Sepsis is a heterogeneous disease, including a wide possible range of patient conditions, pre-existing comorbidities, severity levels, infection incubation times, and underlying immune states. Many investigators have hypothesized that molecular profiling of the host response may better predict sepsis outcomes. Here, we extensively assessed the predictive performance of whole-blood gene expression using a community-based modeling approach. This approach was designed to evaluate predictive capabilities in a manner that was independent of specific methodological preferences, and instead created robust prognostic models across a broad solution space. We developed four state-of-the-art data-driven prognostic models using a comprehensive survey of available data including 21 different sepsis cohorts (both community-acquired and hospital-acquired, $N = 1113$ patients),

**Table 3 Continuous net reclassification index for gene scores over clinical severity scores**

| A | Duke | | | | Sage LR | | | | Sage RF | | | | Stanford | | | |
|---|---|---|---|---|---|---|---|---|---|---|---|---|---|---|---|---|
| | NRI | 95% CI lo | 95% CI hi | P value | NRI | 95% CI lo | 95% CI hi | P value | NRI | 95% CI lo | 95% CI hi | P value | NRI | 95% CI lo | 95% CI hi | P value |
| **Discovery datasets** | | | | | | | | | | | | | | | | |
| EMEXP3850 | 0.095 | -0.309 | 0.499 | 0.646 | -0.305 | -0.709 | 0.099 | 0.139 | 0.105 | -0.484 | 0.695 | 0.726 | 0.2 | -0.207 | 0.607 | 0.335 |
| EMTAB1548 | 0.918 | 0.481 | 1.355 | **3.78E-05** | 0.965 | 0.543 | 1.387 | **7.41E-06** | 2 | 2 | 2 | **0.000** | 0.918 | 0.481 | 1.355 | **3.78E-05** |
| GSE10474 | -0.051 | -0.776 | 0.675 | 0.891 | 0.828 | 0.142 | 1.514 | **0.018** | 2 | 2 | 2 | **0.000** | 0.717 | 0.014 | 1.42 | **0.046** |
| GSE27131 | -0.5 | -1.387 | 0.387 | 0.269 | 0.2 | -1.23 | 1.63 | 0.784 | -0.6 | -2 | 0.953 | 0.449 | 0 | -1.493 | 1.493 | 1 |
| GSE32707 | -0.091 | -0.682 | 0.5 | 0.763 | 0.789 | 0.245 | 1.334 | **0.004** | 1.753 | 1.471 | 2 | **0.000** | 0.896 | 0.364 | 1.427 | **0.001** |
| GSE63042 | 0.343 | -0.136 | 0.821 | 0.161 | 0.856 | 0.404 | 1.308 | **0.000** | 2 | 2 | 2 | **0.000** | 0.706 | 0.246 | 1.167 | **0.003** |
| GSE66099 | 0.915 | 0.567 | 1.264 | **2.66E-07** | 1.183 | 0.834 | 1.532 | **2.99E-11** | 2 | 2 | 2 | **0.000** | 1.022 | 0.69 | 1.354 | **1.64E-09** |
| GSE66890 | 0.774 | 0.208 | 1.34 | **0.007** | 0.249 | -0.313 | 0.811 | 0.385 | 2 | 2 | 2 | **0.000** | 0.821 | 0.259 | 1.383 | **0.004** |
| **Validation datasets** | | | | | | | | | | | | | | | | |
| EMTAB4421 | 0.762 | -0.06 | 1.584 | 0.069 | 0.629 | -0.204 | 1.462 | 0.139 | 0.19 | -0.684 | 1.065 | 0.670 | 0.476 | -0.399 | 1.351 | 0.286 |
| GSE21802 | 0.167 | -0.969 | 1.302 | 0.774 | 0.167 | -0.969 | 1.302 | 0.774 | -0.333 | -1.57 | 0.903 | 0.597 | -0.167 | -1.302 | 0.969 | 0.774 |
| GSE54514 | 1.47 | 0.975 | 1.966 | **6.08E-09** | 0.496 | -0.244 | 1.236 | 0.189 | 1.239 | 0.706 | 1.773 | **5.35E-06** | 0.94 | 0.291 | 1.589 | **0.005** |
| **HAI datasets** | | | | | | | | | | | | | | | | |
| Glue Burns D1-D30 | 0.958 | 0.185 | 1.732 | **0.015** | 0.658 | -0.124 | 1.441 | 0.099 | 0.275 | -0.552 | 1.102 | 0.515 | 0.992 | 0.36 | 1.623 | **0.002** |
| Glue Trauma D1-D30 | 2 | 2 | 2 | **0.000** | 1.958 | 1.878 | 2.039 | **0.000** | 1.333 | 1.067 | 1.6 | **1.15E-22** | 2 | 2 | 2 | **0.000** |
| UF P50 12H | -0.315 | -1.199 | 0.568 | 0.484 | -0.685 | -1.569 | 0.2 | 0.129 | 0.479 | -0.262 | 1.22 | 0.205 | 0.103 | -0.786 | 0.992 | 0.820 |

| B | Duke | | | | Sage LR | | | | Sage RF | | | | Stanford | | | |
|---|---|---|---|---|---|---|---|---|---|---|---|---|---|---|---|---|
| | Mean | Std. Dev | Count p<0.05 | % p<0.05 | Mean | Std. Dev | Count p<0.05 | % p<0.05 | Mean | Std. Dev | Count p<0.05 | % p<0.05 | Mean | Std. Dev | Count p<0.05 | % p<0.05 |
| **Data Interpretation** | | | | | | | | | | | | | | | | |
| All datasets | 0.532 | 0.681 | 6 | 43 | 0.571 | 0.622 | 6 | 43 | 1.032 | 0.938 | 8 | 57 | 0.687 | 0.530 | 9 | 64 |
| Discovery only | 0.300 | 0.493 | 3 | 38 | 0.596 | 0.465 | 5 | 63 | 1.407 | 0.975 | 6 | 75 | 0.660 | 0.341 | 6 | 75 |
| Validation + HAI | 0.840 | 0.769 | 3 | 50 | 0.537 | 0.783 | 1 | 17 | 0.531 | 0.588 | 2 | 33 | 0.724 | 0.706 | 3 | 50 |

A: NRI, confidence intervals, and P-values for mortality prediction for each of the four gene scores over clinical severity scores alone. B: Summary statistics for aggregate samples, broken up by data type (discovery, validation, HAI). NRI, continuous net reclassification index. CI, confidence interval; HAI, hospital-acquired infection. Bold values indicate $p < 0.05$.

with summary AUROCs around 0.85 for predicting 30-day mortality. We also showed that combining the gene-expression-based models with clinical severity scores leads to significant improvement in the ability to predict 30-day mortality, indicating clinical utility.

Prediction of outcomes up to 30 days after the time of sampling represents a difficult task, given that the models must account for all interventions that occur as part of the disease course. An accuracy of 100% is likely not only unachievable but also undesirable, as it would suggest that mortality is pre-determined and independent of clinical care. Given this background, and since similar prognostic power was observed across all individual models and the ensemble model, our prognostic accuracy may represent an upper bound on transcriptomic-based prediction of

| Table 4 Genomic predictors of sepsis mortality | | |
|---|---|---|
| **Model name** | **Direction of change in patients with mortality** | **Genomic features** |
| Duke | Up (5 genes) | *TRIB1, CKS2, MKI67, POLD3, PLK1* |
| | Down (13 genes) | *TGFBI, LY86, CST3, CBFA2T3, RCBTB2, TST, CX3CR1, CD5, MTMR11, CLEC10A, EMR3, DHRS7B, CEACAM8* |
| Sage LR | Up (9 genes) | *CFD, DDIT4, DEFA4, IFI27, IL1R2, IL8, MAFF, OCLN, RGS1* |
| | Down (9 genes) | *AIM2, APH1A, CCR2, EIF5A, GSTM1, HIST1H3H, NT5E, RAB40B, VNN3* |
| Sage RF | Up (13 genes) | *B4GALT4, BPI, CD24, CEP55, CTSG, DDIT4, G0S2, MPO, MT1G, NDUFV2, PAM, PSMA6, SEPP1* |
| | Down (4 genes) | *ABCB4, CTSS, IKZF2, NT5E* |
| Stanford | Up (8 genes) | *DEFA4, CD163, PER1, RGS1, HIF1A, SEPP1, C11orf74, CIT* |
| | Down (4 genes) | *LY86, TST, OR52R1, KCNJ2* |

sepsis outcomes. In addition, since prognostic accuracy was retained across broad clinical phenotypes (children and adults, with bacterial and viral sepsis, with community-acquired and HAIs, from multiple institutions around the world) the models appear to have successfully incorporated the broad clinical heterogeneity of sepsis. The derived discriminatory power of the gene models (AUCs near 0.85) is at least similar to the AUC of proadrenomedullin (0.83) in a recent large prospective trial (TRIAGE study)[8]. Furthermore, the impact of the addition of the severity score to clinical practice could be substantial. If envisioned as a rule-out test for mortality (e.g. setting the threshold at a 95% sensitivity), the Duke and Stanford scores showed large increases in specificity (13–21 percentage point absolute increase) compared with standard clinical severity scores alone. However, peptide assays have the significant advantage of potentially very rapid turnaround times. Moreover, a paucity of randomized data in application of existing biomarkers makes it unclear whether improved risk stratification will actually improve health and/or reduce costs[9].

Sepsis remains difficult to define. The most recent definition of sepsis (Sepsis-3) requires the presence organ dysfunction as measured by an increase in SOFA of two or more points over baseline[1]. Determining the SOFA score can help guide which organ systems are dysfunctional, but this fails to characterize the biological changes are driving the septic response. Molecular tools like the ones developed here provide an opportunity to provide a simple, informative prognosis for sepsis by improving patient risk stratification. Host-response profiles could also help to classify patients with sepsis as opposed to non-septic acute infections. Identifying such high-risk patients may also lead to greater success in clinical trials through improved enrichment strategies. This identification of subgroups or 'endotypes' of sepsis has already been successfully applied to both pediatric and adult sepsis populations[14,15].

The goal of this study was to generate predictive models but not necessarily to define sepsis pathophysiology. However, our community approach identified a large number of genes associated with sepsis mortality that may point to underlying biology. The association with immature neutrophils and inflammation in sepsis has been previously shown[32]. Results of this study confirm this finding as we note increases in the neutrophil chemoattractant IL-8 as well as neutrophil-related antimicrobial proteins (*DEFA4, BPI, CTSG, MPO*). These azurophilic granule proteases may indicate the presence of very immature neutrophils (metamyelocytes) in the blood[33]. Many of these genes have also been noted in the activation of neutrophil extracellular traps (NETs)[34,35]. NET activation leads to NETosis, a form of neutrophil cell death that can harm the host[35]. Whether these involved genes are themselves harmful or are markers of a broader pathway is unknown. Along with immune-related changes, there are changes

in genes related to hypoxia and energy metabolism (*HIF1A, NDUFV2, TRIB1*). Of particular interest is the increase in *HIF1A*, a hypoxia-induced transcription factor. This upregulation is corroborated by previous findings in patients with higher early mortality in the larger E-MTAB-4421.51 cohort[13]. This may be evidence of either a worsening cytopathic hypoxia in septic patients who progress to mortality, or of a shift away from oxidative metabolism ("pseudo-Warburg" effect), or both[36]. Modification of the Warburg effect due to sepsis has been implicated in immune activation[37], trained immunity[38], and immunoparalysis[39].

The present study has several limitations. First, as a retrospective study of primarily publically available data, we are not able to control for demographics, infection, patient severity, or individual treatment. However, our successful representation of this heterogeneity likely contributed to the successful validation in external community-acquired and hospital-acquired sepsis cohorts. Second, despite a large amount of validation data, we do not present the results of any prospective clinical studies of these biomarkers. Prospective analysis will be paramount in translating the test to a clinically relevant assay. In addition, while some rapid PCR techniques could bring the potential turnaround time of a gene-expression-based assay to under 30 min, this will require a substantial engineering effort. Third, the genes identified here were specifically chosen for their performance as biomarkers, not based on known relevance to the underlying pathophysiology of mortality in sepsis. As such, the biological insights gained from these biomarkers will need to be confirmed and expanded on by studies focused on the entire perturbation of the transcriptome during sepsis and through targeted study of individual genes and pathways. Fourth, the use of 30-day mortality as our endpoint is a crude measure of severity, and may miss important intermediate endpoints such as prolonged ICU stay or poor functional recovery. While such intermediate outcomes were not available in the current data, the models' abilities to predict these functional outcomes will need to be tested prospectively. Fifth, despite a seemingly large total N (1113), we were unable to perform robust subgroup analyses (such as infection site or pathogen type), although a broad range of clinical circumstances is included across the datasets. Finally, we note that some may find as a weakness the limited overlap in genes chosen by the four models. However, in the search for sparse models using highly collinear data such as gene expression, near-random selection of variables can occur[40]. The similar performance of the classifiers using disparate gene sets is thus further evidence that these models may be near an upper bound of discriminatory ability using whole-blood gene expression data.

Researchers, clinicians, funding agencies, and the public are all advocating for improved platforms and policies that encourage sharing of clinical trial data[41]. Meta-analysis of multiple studies

leads to results that are more reproducible than from similarly powered individual cohorts[42]. The community approach used here has shown that aggregated transcriptomic data can be used to define novel prognostic models in sepsis. This collaboration of multidisciplinary teams of experts encompassed both analytical and statistical rigor along with deep understandings of both the transcriptomics data and clinical data. To advance beyond the work presented here, more data must be made available, including demographics, treatments, and clinical outcomes, as well as other data types like proteomics and metabolomics. Data-driven collaborative modeling approaches using these data can be effective in discovering new clinical tools.

We have shown comprehensively that patients with sepsis can be risk-stratified based on their gene expression profiles at the time of diagnosis. The overall performance of expression-based predictors paired with clinical severity scores was significantly higher than clinical scores alone in multiple cohorts with heterogeneous sepsis. These gene expression models reflect a patient's underlying biological response state and could potentially serve as a valuable clinical assay for prognosis and for defining the host dysfunction responsible for sepsis. These results serve as a benchmark for future prognostic model development and as a rich source of information that can be mined for additional insights. Improved methods for risk stratification would allow for better resource allocation in hospitals and for prognostic enrichment in clinical trials of sepsis interventions (removing those patients who will likely survive regardless of intervention). Ultimately, prospective clinical trials will be needed to confirm and extend the findings presented here.

## Methods

**Systematic search**. Two public gene expression repositories (NCBI GEO, EMBL-EBI ArrayExpress) were searched for all clinical-gene expression microarray or next-generation sequencing (NGS/RNAseq) datasets that matched any of the following search terms: sepsis, SIRS, trauma, shock, surgery, infection, pneumonia, critical, ICU, inflammatory, nosocomial. Clinical studies of acute infection and/or sepsis using whole blood were retained. Datasets that utilized endotoxin or lipo-polysaccharide infusion as a model for inflammation or sepsis were excluded. Datasets derived from sorted cells (e.g., monocytes, neutrophils) were also excluded.

Overall, 16 studies containing 17 different cohorts were included (Table 1a, b). These 16 studies include expression profiles from both adult[15,17,19,43–52] and pediatric[48,53–56] cohorts. In these cases, the gene expression data were publicly available. When mortality and severity phenotypes were unavailable in the public data, the data contributors were contacted for this information. This included datasets E-MTAB-1548 (refs. [13,57]), GSE10474 (ref. [44]), GSE21802 (ref. [50]), GSE32707 (ref. [47]), GSE33341 (ref. [51]), GSE63042 (ref. [19]), GSE63990 (ref. [52]), GSE66099 (ref. [56]), and GSE66890 (ref. [49]). Furthermore, where longitudinal data were available for patients admitted with sepsis, we only included data derived from the first 48 h after admission. The E-MTAB-4421 and E-MTAB-4451 cohorts both came from the GAinS study[15], used the same inclusion/exclusion criteria, and were processed on the same microarray type. Thus, after re-normalizing from raw data, we used ComBat normalization[58] to co-normalize these two cohorts into a single cohort, which we refer to as E-MTAB-4421.51. For this study, data were included only for patients sampled on the day of hospital admission. In addition to the above 17 datasets, we identified four additional privately held datasets (Table 1c) representing patients with HAI. In-depth summaries of each HAI cohort can be found in the supplementary text.

We selected cohorts as either discovery or validation based on their availability. Studies for which outcome data were readily available were included as discovery cohorts. Only GSE54514 (ref. [17]) was initially held out for validation given its large size and representative patient characteristics. After we had trained the models some outcomes data became newly available, so these were added as validation cohorts[15,50–52]. Additionally, given the known differences in sepsis pathophysiology and gene expression profiles as compared to patients with community-acquired sepsis[56,59], the HAI datasets were set aside as a second validation cohort. The validation cohorts were not matched to the discovery cohort on any particular criteria but rather provide a validation opportunity across a heterogeneous range of clinical scenarios.

**Gene expression normalization**. All Affymetrix datasets were downloaded as CEL files and re-normalized using the gcRMA method (R package affy[60]). Output from other array types were normal-exponential background corrected and then between-arrays quantile normalized (R package limma[61]). For all gene analyses, the mean of probes for common genes was set as the gene expression level. All probe-to-gene mappings were downloaded from GEO from the most current SOFT files.

Two of the cohorts, CAPSOD[19] and the Duke HAI cohort, were assayed via NGS. For compatibility with microarray studies, expression from NGS datasets were downloaded as counts per million total reads (CPM) and were normalized using a weighted linear regression model using the voom method[62] (R package limma[61]). The estimated precision weights of each observation were then multiplied with the corresponding log2(CPM) to yield final gene expression values.

**Prediction models**. Prediction models were built by comparing patients who died within 30 days of hospital admission with sepsis to patients who did not. In the CAPSOD dataset (which was used in model training) we excluded two patients with unclear mortality outcomes, and one patient who died in-hospital but after 30 days. Mortality was modeled as a binary variable as since time-to-event data were not available. Overall, a total of four prognostic models were built by three different academic groups (Duke University, Sage Bionetworks, and Stanford University). All four models started with the same gene expression data in the discovery phase. Each model was built in two phases: a feature selection phase based on statistical thresholds for differential gene expression across all discovery cohorts, and then a model construction phase optimizing classification power. Full descriptions of the four models can be found in the supplementary text and in Supplementary Figs. 1–3.

**Comparison with severity scores**. We compared the prognostic accuracy of the gene scores with the prognostic accuracy of clinical severity scores (APACHE II, PELOD, PRISM, SAPS II, SOFA, and the Denver score) where such information was available. No datasets had more than one clinical severity score type available. These clinical severity scores were not necessarily built to predict mortality in the specific populations in which they were used here, but nonetheless serve as important comparators for the gene expression models. To compare prognostic power in the datasets which included subject-level severity data, LR was performed to predict mortality using either the clinical severity score or the given gene model's output score. We then tested a joint model (mortality as a function of clinical severity and gene score, without interaction term) and measured the AUROC of the combined model. Comparisons were made between AUROCs with paired t-tests. We further computed cNRI index to quantify how well our joint model reclassifies over clinical severity scores alone[63]. The cNRI is the sum of two scores: the improvement in classification of a positive event (here, mortality) by the tested model, plus the improvement in classification of a negative event (here, survival) by the tested model. Each improvement has a possible range of [−1, 1], so the full cNRI has a possible range of [−2, 2]. A score of −2 would mean that every prediction is made worse by the addition of the tested model; a score of 2 means that every prediction is made more accurate by the addition of the tested model. Finally, we calculated test characteristics at both a high-sensitivity cutoff and a high-specificity cutoff, for both clinical scores and gene scores separately, and for the joint clinical-gene models. These are reported as mean ± standard deviation across datasets in summary tables.

**Discriminatory power analyses**. We examined class discriminatory power for separating survivors from non-survivors using ROC curves of the gene scores within datasets. The area under the ROC curves (AUROC) was calculated using the trapezoidal method. Summary ROC curves were calculated via the method of Kester and Buntinx[64]. We examined the ability of the models to predict non-survivors using precision–recall curves generated from the gene scores in each examined dataset. Precision–recall curves of the gene scores were constructed within datasets, and the AUPRC)was calculated using the trapezoidal method.

**Enrichment analysis**. We conducted two analyses to evaluate the functional enrichment of the genes selected as predictors by the four models. This included a targeted enrichment analysis for cell types as previously described[56] and an exploratory enrichment analysis that assessed a large number of functionally annotated gene sets.

In a mixed tissue such as blood, shifts in gene expression can be caused by changes in cell-type distribution. To check for this effect, we used gene expression profiles derived from known sorted cell types to determine whether a given set of genes is enriched for genes represented in a specific cell type. In each curated cell-type vector, a 'score' is calculated by the geometric mean of the upregulated genes minus the geometric mean of the downregulated genes. A higher 'score' represents a greater presence of the given cell type in the differential gene expression signature.

For exploratory enrichment, we curated thousands of gene sets from three widely used databases: gene ontology (GO)[65], the Reactome database of pathways and reactions in human biology[66], and the Kyoto Encyclopedia of Genes and Genomes (KEGG)[67]. Our 12 discovery cohorts had approximately 6000 genes in common, which formed a 'background' set of genes. Genes that are present in the GO/Reactome/KEGG sets but not in the background sets were removed prior to enrichment. We then retained all GO/Reactome/KEGG gene sets containing at least 10% and at least three genes overlapping with the predictor genes. The

remaining GO/Reactome/KEGG gene sets were removed to reduce the multiple testing burden. Exploratory enrichment in each of the curated reference gene sets was performed using two different methodologies: gene-based Fisher's exact test (FET), and, using discovery datasets, expression-based gene set enrichment analysis (GSEA) using GSVA package from bioconductor[68]. Significantly enriched reference gene sets were discovered after adjusting the nominal $P$-values using the Benjamini–Hochberg method.

**Statistics, normalized data and code availability**. All computation and calculations were carried out in the R language for statistical computing (version 3.2.0) and Matlab R 2016a (The MathWorks, Inc.). Significance levels for $P$-values were set at 0.05 and analyses were two-tailed. Analysis source code, final sample scores for the four models along with other relevant analysis results are made available through Synapse, an open source collaborative research platform[69].

**Data availability**. All the raw and normalized gene expression data, mortality and/ or clinical outcomes data, results are made available through Synapse[69]. Readers may access these data for independent research provided they (i) register onto Synapse and (ii) agree to properly acknowledge both the data contributor(s) and the synapse portal as described on the Data Use Requirements page[69].

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

## Acknowledgements

We would like to thank the authors who contributed gene expression data to the public domain that we re-analyzed here. We thank Dr. Michael Bauer for helpful discussion of his group's sepsis data. We thank the Glue Grant investigators for sharing their data publically; they are supported in this by NIGMS Glue Grant Legacy Award R24GM102656. J.F.B.-M., R.A., and E.T. were supported by Instituto de Salud Carlos III (grants EMER07/050, PI13/02110, PI16/01156). R.J.L. was supported by the National Center for Advancing Translational Sciences of the National Institutes of Health under award number UL1TR001417. The CAPSOD study was supported by NIH (U01AI066569, P20RR016480, HHSN266200400064C). P.K. is supported by grants from Bill Melinda Gates Foundation, R01 AI125197-01, 1U19AI109662, and U19AI057229, outside the submitted work. The GAinS study was supported by the National Institute for Health Research through the Comprehensive Clinical Research Network for patient recruitment; Wellcome Trust (Grants 074318 [to J.C.K.], and 090532/Z/09/Z [core facilities Wellcome Trust Centre for Human Genetics including High-Throughput Genomics Group]); European Research Council under the European Union's Seventh Framework Programme (FP7/2007–2013)/ERC Grant agreement no. 281824 (to J.C.K.), the Medical Research Council (98082 [to J.C.K.]); UK Intensive Care Society; and NIHR Oxford Biomedical Research Centre. The Duke HAI study was supported by a research agreement between Duke University and Novartis Vaccines and Diagnostics, Inc. According to the terms of the agreement, representatives of the sponsor had an opportunity to review and comment on a draft of the manuscript. The authors had full control of the analyses, the preparation of the manuscript, and the decision to submit the manuscript for publication. For the University of Florida 'P50' Study, data were obtained from the Sepsis and Critically Illness Research Center (SCIRC) at the University of Florida College of Medicine, which is supported in part by NIGMS P50 GM111152. This work was supported by Defense Advanced Research Projects Agency and the Army Research Office through Grant W911NF-15-1-0107. The views expressed are those of the author and do not reflect the official policy or position of the Department of Veterans Affairs, the Department of Defense or the U.S. Government.

## Author contributions

Study conception and design: T.E.S., T.M.P., R.H., L.O., P.K., E.L.T., L.M.M., and R.J.L.; contributed materials: M.N., J.A.H., A.C., J.F.B.M., R.A., E.T., E.E.D., K.L.B., C.J.H., J.C.K., C.W.W., S.F.K., G.S.G., H.R.W., G.P.P., B.T., L.L.M., F.E.M., E.L.T., and R.J.L.; performed the analyses: T.E.S., T.M.P., and R.H.; drafted manuscript: T.E.S. and T.M.P.; critical revision: all authors.

## Additional information

**Competing interests:** The 'Duke' 18-gene score is the subject of a provisional patent filed by Duke University. The 'Stanford' 12-gene score is the subject of a provisional patent filed by Stanford University. T.E.S. and P.K. are co-founders of Inflammatix, Inc., which has a commercial interest in the 'Stanford' 12-gene score. The remaining authors declare no competing financial interests.

