## [Peer Review File · Nature Communications]

Reviewers' comments:

Reviewer #1 (Remarks to the Author):

Interesting report if the criteria for publishing is „retrospective pilot study of a plethora of molecular markers prone for many biases and limitations including multiple testings“

This bias of this reviewer is to be a clinician emphasising the need for rapid simple aid with biomarkers for triage decisions on the ER. Based on the daily needs of the reviewer and the scientific experience in this respect, the following consideration of major limitation might further improve the paper

- Yes, we know that the clinical estimates for prognosis are very limited. Tools for risk stratification do not only include clinical severity scores such as APACHE or SOFA as well as blood lactate levels. Namely several biomarkers have been proposed. Therefore, to put the current genetic data into research „state of the art“ perspective, comparisons with these biomarkers should be done (ideally individual data directly compared in the results section) or at least the respective AUCs discussed with data of other „biomarkers“ in the literature (e.g. reviewed in Schuetz P., et al Biomarker-guided Personalized Emergency Medicine for All – Hope for another Hype ?. Swiss Med Wkly 2015 ;145:w14079 or Rast A.C., et al. Clinical scores and blood biomarkers for early risk assessment of patients presenting to the emergency department – Critical review. OA Emergency Medicine 2014; 2:1-9)

- Try to apply net-reclassification statistics in a way that it better examines true clinical utility (e.g. like in Kutz A., et al. The TRIAGE-ProADM score for an early risk stratification of medical patients in the emergency department - Development based on a multi-national, prospective, observational study. PLoS ONE 2016; 11: e0168076 DOI:10.1371/journal.pone.0168076

- Discussion: "We developed four state-of-the-art data-driven prognostic models using a comprehensive survey of available data including 21 different sepsis cohorts (both community-acquired and hospital-acquired, N=1,113 patients), with summary AUROCs around 0.85 for predicting 30-day mortality." Although the number of discovery and validation cohorts, respectively seem impressive at first, the relatively small total N limits external validity, namely for clinically important subanalyses. Sepsis is merely a syndrome of heterogenous (sites and causes) of infection, as the authors state in the beginning of the discussion. To estimate clinical relevance the principal causes (i.e. pulmonary, urogenital and abdominal) causes should be subanalysed - microbiological subanalyses would further strengthen the data (e.g. blood culture positive / negative, antibiotic pretreatment vs naive)

- What was the course of the prognostic biomarkers during hospital stay and/or follow-up (i.e. decreasing with favourable outcome and vice versa?)

- For emergency care and triage decisions timing is everything (minutes is better than hours and far better than days...). So what's the current estimated turn-around time . . . estimate on whether a (needed) turnaround time of minutes will be ever feasible to become accepted for clinical routine

- Define specifically which sepsis definitions were used to classify the patients in the validation cohorts (the recently published sepsis III criteria, where basically the old „severe sepsis“ and becomes termed „sepsis“, thus the definition is more stringent than the old one). Were all the diagnostic criteria similar in all validation cohorts ?

Reviewer #2 (Remarks to the Author):

This study provides some convincing evidence that gene expression models can predict the outcome of sepsis patients. My main concern with the manuscript is that I would like to see a more detailed discussion of the practical significance of the findings. Specifically, I would like to know how much the proposed methodology improved prediction accuracy compared to a model that uses only standard clinical severity predictors. The authors consider this question to a degree, but they merely state that the difference in the AUROC between a model using only standard

predictors and a model using standard predictors plus genetic predictors was statistically significant. However, they do not state how large this difference is. More importantly, the difference in the AUROC does not readily translate to practical significance. I would be more interested in a more basic question: how many more sepsis patients are correctly classified when genetic factors are considered when predicting risk?

AUROC gives a measure of the overall predictive accuracy of a model across a variety of cutoffs. However, in practice, one would have to choose a specific cutoff in order to use the model. Thus, I would be interested in seeing how these models perform for some specific choices of the cutoff. The authors could use a cutoff to set the specificity to be 90% (as they did for Supplementary Figure 5) or some other cutoff. Indeed, it would potentially be interesting to see the results for several possible cutoffs. For each cutoff, I would like to see (in a supplementary table) the confusion matrix for each testing/HAI data set for each of the following models: 1) gene expression alone, 2) standard predictors alone, 3) gene expression plus standard predictors, 4) baseline error model (all patients are assigned to the largest class, which is usually the "survived" class). In addition, I would like to see a figure similar to Supplementary Figure 5 for each of the four models listed above (rather than only for the gene expression model). I would also like to know the difference between models 2) and 3) with respect to each of the five measures in Supplementary Figure 5. In other words, how much does accuracy, sensitivity, specificity, etc. increase when gene expression is included in the model (compared to a model that uses only standard predictors).

A few other (more minor) comments:

On lines 279-283, the authors write, "The AUPRCs for non-survivor prediction were notably lower than the AUROCs, suggesting that the models' primary utility may be in ruling out mortality for individuals much less likely to die within 30 days (those less likely to require substantial intervention) as opposed to accurately identifying the minority of patients who are highly likely to die within 30 days." I am not sure why that statement follows from the fact that the AUPRCs were lower than the AUROCs. We would expect the AUPRCs to be lower since each of the validation data sets are highly unbalanced (with more survivors than deaths). The AUROC is always bounded below by 0.5, whereas the AUPRC is bounded below by $D/(D+S)$, D and S are the number of deaths and survivors, respectively. Given that S is significantly larger than D for each validation set, it is not surprising that the AUPRCs are lower than the AUROCs.

I will note that some authors have suggested that the AUPRC is a better metric than AUROC when the data is unbalanced, as is the case for the validation data sets in this study. (See for example Saito and Rehmsmeier, *PLoS One*. 2015; 10(3): e0118432.) I will let the authors decide if greater emphasis should be placed on the AUPRC in this study.

I was also confused as to how the AUROC was calculated for the random forest model. The standard random forest algorithm assigns a predicted class for each observation by taking the majority vote of the individual trees. Thus, there is only a single binary prediction, not a continuous risk score that would allow one to choose a cutoff. One could potentially generate such a risk score by calculating the percentage of trees that predicted a death rather than using a simple majority vote. If that is the case, that should be spelled out clearly in the description of the method.

Finally, I would encourage the authors to make their R source code publicly available to ensure that their work is reproducible.

Reviewer #3 (Remarks to the Author):

The authors performed a meta-analysis of selected previously published datasets pertaining to

critical illness due to all causes of sepsis, non-septic admissions such as burns and traumatic injury as well as pediatric intensive care patients. Using standard methods in cross-platform normalization, batch corrections and statistical models, the authors' aim was to derive a molecular predictor of 30-day mortality (handled in binary fashion). To this aim, the authors resorted to constructing no less than four statistical models, which according to the understanding of this reviewer were based on the center responsible for the analysis (Duke, Sage Bionetworks and Stanford). The accuracy of the derived putative molecular predictors was subsequently assessed in selected validation cohorts. Moreover, identified genes per model were combined and further refined to an "ensemble" gene set and proposed as a 30-day mortality predictor. Model performances were primarily assessed by ROC AUCs and AUPRCs and subsequently gene sets were compared to clinical scoring systems to assess predictive performances in isolation or in combinations thereof.

Comments

(1) While the research question is an important one, there is a shortage in reporting specific methodologies and several components of the analysis, interpretation and conclusions deserve clarification. Firstly, the lack of overlap in gene sets identified as 30-day mortality predictors per statistical model is a major concern. The NT5E, DEFA4, RGS1 and SEPP1 genes were seemingly an exception; albeit also poorly overlapping across models. Not one gene was consistently identified across models and, not surprisingly, the authors found poor correlation between models. Although this may indicate unaddressed bias in each of the evaluated models, this also brings the methodology in combining these datasets (lacking in specifics) into question. For example, what was the distribution of principal components before and after normalization, SVA adjustment etc..? The reported predictive performances (AUROC and AUPRC) have predominantly low AUCs, very broad confidence margins and were not found to possess independent prognostic value as compared to clinically-derived scores.

(2) The authors went on to reportedly combine gene expression (molecular) and clinical scores into a joint mortality prediction model. Again, specifics in methodology is lacking with only a reported probability as result. What were the AUROC and confidence intervals of the clinical scores in isolation? The supposed improvement over clinical scores alone in mortality prediction may be based on an analysis that is not a level playing field, in that for example APACHE II scores were not built on these datasets. A customized clinical risk prediction model may have performed better, and the improvement in prediction may have been more modest.

(3) Insufficient detail is included in the manuscript to describe combining publicly available datasets from different platforms, predictive performances considering molecular and clinical models as well as pathway analyses. Can the authors please include more information?

(4) In their pathway analysis, why have the authors chosen for targeted enrichment analysis for cell types using data from reference 38? In silico deconvolution of gene expression to specific peripheral blood cell types has been applied many times using data from benchmark studies, for example Abbas et al 2009, and more recently Newman et al 2015? Have the authors considered other pathway analysis suites such as ingenuity and/or GSEA? These represent highly curated databases and it is unclear to this reviewer why the authors adopted the approach reported in their study.

(5) In their comparison to severity scores have the authors assessed all scores in their joint model, that is, APACHE II, PELOD, PRISM, SAPS II, SOFA, and the Denver scores? Where they evaluated independently? The authors should provide more extensive data and methodology to support their claim that "the gene expression-based predictors add significant prognostic utility to standard clinical metrics". Simply reporting a probability is not sufficient.

(6) Line 410, please correct the reference citation style to the journal requirements

Reviewer #1 (Remarks to the Author):

Interesting report if the criteria for publishing is „retrospective pilot study of a plethora of molecular markers prone for many biases and limitations including multiple testings“ This bias of this reviewer is to be a clinician emphasising the need for rapid simple aid with biomarkers for triage decisions on the ER. Based on the daily needs of the reviewer and the scientific experience in this respect, the following consideration of major limitation might further improve the paper

We are pleased that the reviewer agrees with the need for improved biomarkers for risk stratification in sepsis.

- Yes, we know that the clinical estimates for prognosis are very limited. Tools for risk stratification do not only include clinical severity scores such as APACHE or SOFA as well as blood lactate levels. Namely several biomarkers have been proposed. Therefore, to put the current genetic data into research „state of the art“ perspective, comparisons with these biomarkers should be done (ideally individual data directly compared in the results section) or at least the respective AUCs discussed with data of other „biomarkers“ in the literature (e.g. reviewed in Schuetz P., et al Biomarker-guided Personalized Emergency Medicine for All – Hope for another Hype ?. Swiss Med Wkly 2015 ;145:w14079 or Rast A.C., et al. Clinical scores and blood biomarkers for early risk assessment of patients presenting to the emergency department – Critical review. OA Emergency Medicine 2014; 2:1-9)

The reviewer brings up an excellent point—many other biomarkers for risk stratification have been proposed. Many papers reporting on heavily-studied biomarkers such as proADM, PCT, proET-1, or suPAR have not compared these head-to-head. Similarly, and unfortunately, we do not have any data, let alone sample-level data, on those novel biomarkers here. Such a test will require prospective study. However, to try to answer the reviewer, we examined whether the gene expression levels of these four targets (*ADM*, *CALCA*, *EDN1*, and *PLAUR*, respectively) held any prognostic power in the studied datasets. Not surprisingly, as gene levels often poorly correlate with propeptide levels, the mean AUROC for prediction of mortality was 0.51, 0.48, 0.53, and 0.46 for these genes, respectively. We have not added this to the manuscript as we feel it is a straw-man argument.

We have added the suggested citations to the introduction, ensuring that the import of peptide biomarkers is not overlooked: **“Some peptide markers of sepsis severity have been validated (e.g. proadrenomedullin among others), but these are not yet cleared for clinical use.”**

In addition, we have significantly expanded the discussion section to include a brief review of the summary characteristics of a few of the better-studied alternative biomarkers to allow for easy comparison to our results. The discussion now reads:

“The derived discriminatory power of the gene models (AUCs near 0.85) are at least similar to the AUC of proadrenomedullin (0.83) in a recent large prospective trial (TRIAGE study). However, peptide assays have the significant advantage of potentially very rapid turnaround times. Moreover, a paucity of randomized data in application of existing biomarkers makes it unclear whether improved risk stratification will actually improve health and/or reduce costs.”

- Try to apply net-reclassification statistics in a way that it better examines true clinical utility (e.g. like in Kutz A., et al. The TRIAGE-ProADM score for an early risk stratification of medical patients in the emergency department - Development based on a multi-national, prospective, observational study. PLoS ONE 2016; 11: e0168076 DOI:10.1371/journal.pone.0168076

The reviewer’s request is a clear theme from all reviewers; namely, the question of added utility of a new biomarker vs. standard measures of stratification. The cited work (TRIAGE study) was prospectively designed for such a purpose, and so we cannot be as thorough as that work. In particular, the various

datasets used different clinical risk-stratification scores, preventing us from identifying universal risk bands that could evenly apply across all datasets. Thus, to prevent the appearance of ‘p-hacking’, we have chosen to implement the continuous net reclassification improvement (cNRI) index, which has an absolute range of 0-2, and which can be used to judge the improvement in reclassification of an additional risk score (i.e., the gene model in addition to standard model). Supplementary Table 6 shows cNRI data for all four gene models compared to clinical severity predictors, along with 95% CI and p-values for each dataset. The results have been modified to read:

“We next examined continuous net reclassification improvement (cNRI) index to quantify how well the model with gene scores reclassifies survivors over the model with clinical severity scores in each of these same datasets (Supplementary Table 6). In the validation and HAI cohorts, the mean NRI was 0.53-0.84 (potential range 0-2, where 2 reflects all patients with improved classification). For the Duke and Stanford scores, half of the validation and HAI datasets showed significant NRI compared to standard predictors alone. This suggests that the gene expression-based predictors add significant prognostic utility to standard clinical metrics.

Finally, we examined test characteristics at a high-sensitivity cutoff (95%) and a high-specificity cutoff (95%) for the gene scores in comparison to baseline error models (Supplementary Table 7) and in comparison to clinical severity scores (Supplementary Table 8). Overall mean accuracy of the joint clinical and gene scores was higher in the validation and HAI datasets (0.58-0.72 and 0.64-0.79 across the models, respectively) compared to clinical scores alone (0.57 and 0.62, respectively). “

We note that the cNRI does not produce tables of number of patients reclassified, only an overall estimate of gain in performance. Reporting of a traditional NRI requires quite a large table (e.g., Table 4 (Kutz et al. 2016) has ~60 cells), and here we would need to do this for 4 models for 9 datasets, which would be likely too dense for any reader to glean information. Clearly, as we mention in the discussion, prospective study is needed to fully evaluate the clinical utility of the proposed models. These data merely show the possibility that the gene scores may have clinical utility when tested prospectively.

- Discussion: “We developed four state-of-the-art data-driven prognostic models using a comprehensive survey of available data including 21 different sepsis cohorts (both community-acquired and hospital-acquired, N=1,113 patients), with summary AUROCs around 0.85 for predicting 30-day mortality.” Although the number of discovery and validation cohorts, respectively seem impressive at first, the relatively small total N limits external validity, namely for clinically important subanalyses. Sepsis is merely a syndrome of heterogenous (sites and causes) of infection, as the authors state in the beginning of the discussion. To estimate clinical relevance the principal causes (i.e. pulmonary, urogenital and abdominal) causes should be subanalysed

We see the inclusion of 1,113 patients as relatively large, not relatively small, compared to the average reporting of a new biomarker (but obviously much smaller than large studies of established biomarkers, such as the TRIAGE study). Still, the reviewer’s point is well-taken; sepsis is a highly heterogeneous syndrome, and subgroup analyses are worthy. The datasets included in this study were each largely homogenous examinations of a single clinical setting (e.g., ventilated patients with severe influenza, or post-surgical patients with bacteremia, etc.). Due to technical batch effects between studies, we cannot perform subgroup analyses that would approach statistical significance, and do not want to offer non-robust conclusions. We have added this point to the ‘weaknesses’ paragraph:

“Fifth, despite a seemingly large total N (1,113), we were unable to perform robust subgroup analyses (such as infection site or pathogen type), although a broad range of clinical circumstances is included across the datasets.”

- *microbiological subanalyses would further strengthen the data (e.g. blood culture positive / negative, antibiotic pretreatment vs naive)*

All patients here were defined as having an infection according to the diagnostic criteria set forth in their initial studies. In almost all cases, this was a two-physician retrospective chart review that required microbiological evidence of infection. We do not have data on whether this was positive blood culture (bacteremia) or other positive culture result. The patients at admission to the hospital are assumed to be antibiotic naïve. The hospital-acquired-infection patients were only included if they did not have a prior diagnosis of infection. While broad classes of infectious disease (bacterial, viral, fungal) may induce different host gene expression responses, almost no datasets had more than one class of infection. We have modified Table 1 to add the percent of total subjects that had bacterial infections. We have also added the subgroup analysis limitation statement above to the weaknesses paragraph.

- *What was the course of the prognostic biomarkers during hospital stay and/or follow-up (i.e. decreasing with favourable outcome and vice versa?)*

Only a small subset of studies had longitudinal samples that allowed for this analysis. The data from these studies is now available as Supplemental Figure 6. The results now read:

“We examined the effects of clinical time course on the gene scores in the two validation datasets that tracked longitudinal data (GSE21802 and GSE54154; Supplemental Figure 6). We found no differences in slope (change in score over time) between the survivors and non-survivors, although the scores in non-survivors were significantly higher than in survivors during the entire hospital stay, possibly indicating a failure to restore homeostasis.”

- *For emergency care and triage decisions timing is everything (minutes is better than hours and far better than days...). So what's the current estimated turn-around time . . . estimate on whether a (needed) turnaround time of minutes will be ever feasible to become accepted for clinical routine*

A turnaround time of an hour or less is certainly feasible; some rapid PCR techniques can accomplish amplification in less than 10 minutes (e.g. isothermal techniques). Although work on translation to a potential IVD is underway, we feel the engineering/technical aspects of such a test are outside the scope of the present manuscript. We have added this to the weaknesses paragraph: *“In addition, while some rapid PCR techniques could bring the potential turnaround time of a gene-expression-based assay to under 30 minutes, this will require a substantial engineering effort.”*

- *Define specifically which sepsis definitions were used to classify the patients in the validation cohorts (the recently published sepsis III criteria, where basically the old „severe sepsis“ and becomes termed „sepsis“, thus the definition is more stringent than the old one). Were all the diagnostic criteria similar in all validation cohorts ?*

Diagnostic criteria were, as the reviewer may imagine, not perfectly consistent across the 21 included cohorts. In particular, all of the studies, because they are historical cohorts, were designed prior to Feb 2016, when Sepsis-3 was published. However, we see this as a strength, not a weakness. The host response doesn't know what external labels we place on a patient; we have shown here that the classifier is robust to minor variations in the exact definition of sepsis. In clinical application, of course a prospective study with appropriate, locked inclusion criteria would be needed.

Reviewer #2 (Remarks to the Author):

This study provides some convincing evidence that gene expression models can predict the outcome of sepsis patients. My main concern with the manuscript is that I would like to see a

more detailed discussion of the practical significance of the findings. Specifically, I would like to know how much the proposed methodology improved prediction accuracy compared to a model that uses only standard clinical severity predictors. The authors consider this question to a degree, but they merely state that the difference in the AUROC between a model using only standard predictors and a model using standard predictors plus genetic predictors was statistically significant. However, they do not state how large this difference is. More importantly, the difference in the AUROC does not readily translate to practical significance. I would be more interested in a more basic question: how many more sepsis patients are correctly classified when genetic factors are considered when predicting risk?

We thank the reviewer for the positive comments. We have added the individual differences gained from the joint classifier to the main text; these are merely the subtraction of column 2 from the 'joint' columns of Supplementary Table 5. The text now reads: "each combination significantly outperformed clinical severity scores alone (mean difference Duke 0.077; Sage LR 0.076; Sage RF 0.16; Stanford 0.098; all paired t-tests $p \leq 0.01$)". We have also reworked the presentation of this table for clarity.

In terms of reclassification, we refer the Reviewer to our responses above, namely to the addition of NRI statistics (Supplementary Table 6), and the accompanying changes in the text. Although this comparison has some weaknesses, we hope that it answers the Reviewer's concerns.

AUROC gives a measure of the overall predictive accuracy of a model across a variety of cutoffs. However, in practice, one would have to choose a specific cutoff in order to use the model. Thus, I would be interested in seeing how these models perform for some specific choices of the cutoff. The authors could use a cutoff to set the specificity to be 90% (as they did for Supplementary Figure 5) or some other cutoff. Indeed, it would potentially be interesting to see the results for several possible cutoffs. For each cutoff, I would like to see (in a supplementary table) the confusion matrix for each testing/HAI data set for each of the following models: 1) gene expression alone, 2) standard predictors alone, 3) gene expression plus standard predictors, 4) baseline error model (all patients are assigned to the largest class, which is usually the "survived" class).

The Reviewer is requesting 4 confusion matrices for 4 different scores in at least 9 different cohorts (those with severity data), which would come out to $4 \times 4 \times 9 = 126$ tables for a single cutoff! While this reportage is certainly possible (and we are happy to add it if the reviewer insists), we have instead provided two sets of summary data that we think answer this question. The first table (Supplementary Table 7) shows mean +/- s.d. for test characteristics aggregated across all cohorts for both the gene scores and the baseline error model at two different cutoffs (one with a sensitivity near 95%, the other with a specificity near 95%). The second table (Supplementary Table 8) similarly shows mean +/- s.d. for test characteristics aggregated across the 9 cohorts for which we have severity data. Here we show the test characteristics for the severity score alone, the gene scores alone, and then the joint gene + severity model. It is necessary to repeat the gene-score-only data since this is a different group of datasets than in Supplementary Table 7. The results have now been updated to read:

"Finally, we examined test characteristics at a high-sensitivity cutoff (95%) and a high-specificity cutoff (95%) for the gene scores in comparison to baseline error models (Supplementary Table 7) and in comparison to clinical severity scores (Supplementary Table 8). Overall mean accuracy of the joint clinical and gene scores was higher in the validation and HAI datasets (0.58-0.72 and 0.64-0.79 across the models, respectively) compared to clinical scores alone (0.57 and 0.62, respectively)."

In addition, I would like to see a figure similar to Supplementary Figure 5 for each of the four models listed above (rather than only for the gene expression model). I would also like to know the difference between models 2) and 3) with respect to each of the five measures in

Supplementary Figure 5. In other words, how much does accuracy, sensitivity, specificity, etc. increase when gene expression is included in the model (compared to a model that uses only standard predictors).

We hope that the data listed in the answer to the comment above is sufficient. The addition of three more supplemental figures to this paper, all of which are redundant with new Supplementary Table 8, seems to us only to add complexity without transmitting extra information. We can turn the data from Supplementary Table 8 into new figures if the reviewer and editor wish.

A few other (more minor) comments:

On lines 279-283, the authors write, "The AUPRCs for non-survivor prediction were notably lower than the AUROCs, suggesting that the models' primary utility may be in ruling out mortality for individuals much less likely to die within 30 days (those less likely to require substantial intervention) as opposed to accurately identifying the minority of patients who are highly likely to die within 30 days." I am not sure why that statement follows from the fact that the AUPRCs were lower than the AUROCs. We would expect the AUPRCs to be lower since each of the validation data sets are highly unbalanced (with more survivors than deaths). The AUROC is always bounded below by 0.5, where as the AUPRC is bounded below by $D/(D+S)$, D and S are the number of deaths and survivors, respectively. Given that S is significantly larger than D for each validation set, it is not surprising that the AUPRCs are lower than the AUROCs.

I will note that some authors have suggested that the AUPRC is a better metric than AUROC when the data is unbalanced, as is the case for the validation data sets in this study. (See for example Saito and Rehmsmeier, PLoS One. 2015; 10(3): e0118432.) I will let the authors decide if greater emphasis should be placed on the AUPRC in this study.

We agree with the Reviewer that the AUPRCs were expected to be lower than the AUROCs, but did not wish to highlight this in the manuscript because it would seem to be downplaying negative results. The statement that the models may be better served at identifying low-risk patients as opposed to high-risk patients follows from the unbalanced nature of the data, which is, we agree, a causative factor of the low AUPRCs. To avoid any confusion, we have modified the statement to read: "The AUPRCs for non-survivor prediction were notably lower than the AUROCs, **as was expected from the highly unbalanced classes (rare mortalities). This suggests that** the models' primary utility may be in ruling out mortality for individuals much less likely to die within 30 days".

We chose to highlight AUROCs instead of AUPRCs because the latter measure is not familiar to physicians (as many of our physician co-authors attested). For instance, we could not find any literature reporting AUPRC for other traditional sepsis prognostic markers that would stand as comparators. We have included both measures to ensure that interested parties have sufficient data to understand classifier performance.

I was also confused as to how the AUROC was calculated for the random forest model. The standard random forest algorithm assigns a predicted class for each observation by taking the majority vote of the individual trees. Thus, there is only a single binary prediction, not a continuous risk score that would allow one to choose a cutoff. One could potentially generate such a risk score by calculating the percentage of trees that predicted a death rather than using a simple majority vote. If that is the case, that should be spelled out clearly in the description of the method.

We thank the reviewer for this point. As the Reviewer deduced, indeed the classification probabilities were computed as the proportion of trees that predicted mortality. To reflect this, we added the following paragraph in the Supplemental methods.

"Sage RF model used a penalized classification probability-based random forest (classRF)

algorithm as described in Malley et. al.12. In brief, the classRF algorithm first creates a bootstrap sample set with replacement from the available samples by leaving out a certain percentage of training data. Later a classification tree is built for each bootstrap to the greatest extent possible, but requiring a minimum of 10% of the samples as nodes. Finally, the probability of each sample is calculated as the proportion of predicted non-survivors in the final nodes of all the bootstraps. To reduce the set of features that predicts mortality, penalised classRF is used. “

Finally, I would encourage the authors to make their R source code publicly available to ensure that their work is reproducible.

We appreciate reviewers encouragement to publish the reproducible code. In the view of reproducing our results, we make all the source codes, raw and normalised data along with final classification scores for each sample from all four models, available online through Synapse, an open source collaborative platform (www.synapse.org). To reflect this change we added the following section in the main text:

“To reproduce the discovery of gene sets, all the analysis source code, raw and normalised gene expression data (if available in public domain), and final sample scores for the four models. Code to reproduce the discovery of the gene sets are made available in Synapse, an open source collaborative research platform⁵³”

Reviewer #3 (Remarks to the Author):

The authors performed a meta-analysis of selected previously published datasets pertaining to critical illness due to all causes of sepsis, non-septic admissions such as burns and traumatic injury as well as pediatric intensive care patients. Using standard methods in cross-platform normalization, batch corrections and statistical models, the authors’ aim was to derive a molecular predictor of 30-day mortality (handled in binary fashion). To this aim, the authors resorted to constructing no less than four statistical models, which according to the understanding of this reviewer were based on the center responsible for the analysis (Duke, Sage Bionetworks and Stanford). The accuracy of the derived putative molecular predictors was subsequently assessed in selected validation cohorts. Moreover, identified genes per model were combined and further refined to an “ensemble” gene set and proposed as a 30-day mortality predictor. Model performances were primarily assessed by ROC AUCs and AUPRCs and subsequently gene sets were compared to clinical scoring systems to assess predictive performances in isolation or in combinations thereof.

We thank the reviewer for this accurate summary of the work.

Comments

(1) While the research question is an important one, there is a shortage in reporting specific methodologies and several components of the analysis, interpretation and conclusions deserve clarification. Firstly, the lack of overlap in gene sets identified as 30-day mortality predictors per statistical model is a major concern. The NT5E, DEFA4, RGS1 and SEPP1 genes were seemingly an exception; albeit also poorly overlapping across models. Not one gene was consistently identified across models and, not surprisingly, the authors found poor correlation between models. Although this may indicate unaddressed bias in each of the evaluated models, this also brings the methodology in combining these datasets (lacking in specifics) into question. For example, what was the distribution of principal components before and after normalization, SVa adjustment etc..? The reported predictive performances (AUROC and AUPRC) have predominantly low AUCs, very broad confidence margins and were not found to possess independent prognostic value as compared to clinically-derived scores.

The Reviewer has correctly pointed out that there were few genes chosen in common between the

models. However, the point about combining datasets is a red herring. Only the Sage models (LR/RF) were SVA-normalized during the discovery phase, and none of the methods used pooled normalization methods that would allow for cross-dataset PCA. Instead, in each case, models were jointly optimized across a number of disparate datasets. In order to clarify this, we have modified the methods to reflect these changes.

The Reviewer writes that identifying a small number of genes in common is a major concern, but without justification. Many genes are highly correlated; it is thus the case that disparate modelling techniques that are seeking sparse classification models may choose nearly randomly in removing genes that are highly correlated (e.g., lasso regression, Tibshirani, JRSSB, 1996). Moreover, using highly disparate methodologies of classification was a way of sampling the possible classifier space, but certainly made more likely the possibility of finding disparate models. The advantage of our model is that we provide evidence that the classification power demonstrated by the 4 models in our manuscript is likely at or near the upper bound of the possible discriminatory power available. We thus argue that finding a large number of disparate models with similar discriminatory power should be reassuring, not seen as a weakness, since it is evidence that another group is unlikely to substantially outperform our results merely by selecting some different genes or a different method. We have added this to the discussion:

“Finally, we note that some may find as a weakness the limited overlap in genes chosen by the four models. However, in the search for sparse models using highly collinear data such as gene expression, near-random selection of variables can occur. The similar performance of the classifiers using disparate gene sets is thus further evidence that these models may be near an upper bound of discriminatory model for whole-blood gene expression data.”

The comment regarding predictive performance will be answered in the Reviewer’s point below.

(2) The authors went on to reportedly combine gene expression (molecular) and clinical scores into a joint mortality prediction model. Again, specifics in methodology is lacking with only a reported probability as result. What were the AUROC and confidence intervals of the clinical scores in isolation? The supposed improvement over clinical scores alone in mortality prediction may be based on an analysis that is not a level playing field, in that for example APACHE II scores were not built on these datasets. A customized clinical risk prediction model may have performed better, and the improvement in prediction may have been more modest.

The AUROCs of the clinical scores in isolation are reported in the second column of Supplementary Table 5. These have been broken into Discovery and Validation subgroups for exactly this reason; the ‘fairest’ comparison is the Validation and HAI groups. We concur that customized clinical risk predictors may have better accuracy for that specific cohort than general clinical scores such as APACHE II. However, a score customized for each cohort is by definition not generalized and is unlikely to serve as a useful clinical tool. Regarding improvements in prediction, please see our answers to Reviewer 1 that discuss NRI.

(3) Insufficient detail is included in the manuscript to describe combining publicly available datasets from different platforms, predictive performances considering molecular and clinical models as well as pathway analyses. Can the authors please include more information?

The Supplemental Information contains substantial explanation of how the methods were carried out. While we agree that moving these data to the main text would enhance readability, we are limited by word limits. If the Reviewer has specific methods that were not addressed in our revisions, we would be happy to expand on those specifically. Furthermore, the online methods have been significantly expanded.

(4) In their pathway analysis, why have the authors chosen for targeted enrichment analysis for cell types using data from reference 38? In silico deconvolution of gene expression to specific peripheral blood cell types has been applied many times using data from benchmark studies, for example Abbas et al 2009, and more recently Newman et al 2015? Have the authors considered other pathway analysis suites such as ingenuity and/or GSEA? These represent highly curated databases and it is unclear to this reviewer why the authors adopted the approach reported in their study.

Our approach takes a small number of genes and tries to determine whether their differential expression may be due to cell type shifts. In contrast, cellular deconvolution as described by Abbas et al or Newman et al is a different approach which takes as input an entire gene expression vector and tries to determine the relative presence of all cell types. Deconvolution in this manner would be infeasible for two reasons: first, we would get sample-level or dataset-level outputs across all genes present, instead of learning interesting biology about our particular genes under study. Second, both described methods are heavily influenced by technical effects in their basis matrices, such that their accuracy falls substantially outside of Affymetrix arrays (we have shown this in yet-unpublished data). Thus, the application of these methods would be impossible across datasets. By contrast, by looking at relative differences, we eliminate platform effects, making the present method appropriately suited to the task at hand.

(5) In their comparison to severity scores have the authors assessed all scores in their joint model, that is, APACHE II, PELOD, PRISM, SAPS II, SOFA, and the Denver scores? Where they evaluated independently? The authors should provide more extensive data and methodology to support their claim that “the gene expression-based predictors add significant prognostic utility to standard clinical metrics”. Simply reporting a probability is not sufficient.

We only had one score (APACHE II, PRISM, etc.) per dataset. Each score was evaluated both alone and in combination with the gene score (Supplementary Table 5). In no cohorts did we have sample-level data for two scores, so we could not, as the reviewer suggests, form a joint model of, e.g., SOFA + APACHE II + gene score (we have added this statement to the methods). In terms of reporting methods, they have been modified to read ...

“We further calculated continuous net reclassification scores for the joint model over clinical severity scores alone. Finally, we calculated test characteristics at both a high-sensitivity cutoff and a high-specificity cutoff, for both clinical scores and gene scores separately, and for the joint clinical-gene models. These are reported as mean +/- standard deviation across datasets in summary tables.”

Please see our responses to Reviewer 1 regarding the addition of NRI statistics.

(6) Line 410, please correct the reference citation style to the journal requirements

We thank the reviewer for catching this error; it has been corrected.

Reviewers' comments:

Reviewer #2 (Remarks to the Author):

The authors did an excellent job of responding to my concerns in the earlier critique. I have only a few minor concerns remaining. The authors use the cNRI measure to evaluate how much gene expression can improve classification. I was not familiar with this particular statistic, and it was surprisingly difficult to find information about it via an Internet search. I imagine other readers may be unfamiliar with this statistic, so some additional context and explanation would be helpful. Also, it appears that different sources define this measure differently. (In the first article I found, the cNRI can range between -2 and 2, whereas the others appear to use a different version that ranges only between 0 and 2.)

I recommend that the authors add a few sentence to the Methods explaining (in general terms) how this statistic is calculated and how it should be interpreted. They should also either provide a citation to a more detailed description of the statistic or provide such a description in the Supplementary Information.

Second, I am grateful to the authors for quantifying the improvement that results from incorporating gene expression into the predictive models. However, I would like to see some additional discussion of the significance of these findings. Do the results suggest that gene expression can produce a big enough increase in predictive accuracy to justify using it in the clinic? Or is the improvement too small? If it is too small to be clinically useful, are there other reasons the findings are important? Perhaps they provide some insight into the underlying etiology of sepsis? The paper could be strengthened by discussing the significance/implications of the results in more detail.

Reviewer #3 (Remarks to the Author):

Response to authors' rebuttal:

On comment 1:

The authors have now included further methodological details in the supplement as suggested. However, the lack of overlap between models remains a major concern and major limitation to the study (certainly not only to some). No attempt was made to provide objective arguments and interpretations on their study design as well as analysis other than a certain degree of circular logic. Sure, multicollinearity is a notorious problem in the analysis of high-dimensional data that undoubtedly influences parameter estimation due to inflation of variance in regression parameters. It becomes a severe problem in disparate analysis when a model is trained on a first set of selected studies, and evaluated in another set of selected studies of unknown collinearity structure. The motivation to dismiss as "red herring" the concepts in combining blood genomic datasets for proper meta-analysis of gene expression data in sepsis, which would have allowed for better handling of the differing collinearity structures across different datasets, is unclear and quite subjective. If one considers the authors' stance on the lack of overlap between models, then what is the difference between this study and simply combining previously derived and published gene expression markers for possible prognosis, for example genes and/or gene sets from Wong HR et al. *Physiol Genomics*. 2007;30:146–155 and Dolinay T et al. *Am J Respir Crit Care Med*. 2012;185:1225–1234 and Tsalik EL et al. *Genome Med*. 2014;6:111 and Almansa R et al. *J Infect*. 2015;70:445–456 and Scicluna BP et al. *Am J Respir Crit Care Med*. 2015 Oct 1;192(7):826-35, into one predictive model? Not much. Ultimately, the authors do not provide a potential predictive tool as they sought, but rather showed that by analyzing blood genomic data derived from different studies and analyzed in disparate fashion, using different statistical models and their unaddressed baggage of assumptions on the same discovery/validation sets, revealed substantial discrepancies between models that did not outperform other candidate prognostic biomarkers such

as pro-adrenomedullin. The authors should be more clear about these points in their discussion and conclusion.

On comment 2:

The authors have addressed the comment

On comment 3:

The authors have addressed the comment

On comment 4:

The authors have not addressed a question on their pathway analysis methods. Why haven't the authors considered using highly curated databases for pathway analysis such as GSEA or Ingenuity? The authors should at the least provide pathway enrichment results based on these databases.

On comment 5:

The authors have addressed the comment

Reviewer #2 (Remarks to the Author):

The authors did an excellent job of responding to my concerns in the earlier critique. I have only a few minor concerns remaining. The authors use the cNRI measure to evaluate how much gene expression can improve classification. I was not familiar with this particular statistic, and it was surprisingly difficult to find information about it via an Internet search. I imagine other readers may be unfamiliar with this statistic, so some additional context and explanation would be helpful. Also, it appears that different sources define this measure differently. (In the first article I found, the cNRI can range between -2 and 2, whereas the others appear to use a different version that ranges only between 0 and 2.)

I recommend that the authors add a few sentence to the Methods explaining (in general terms) how this statistic is calculated and how it should be interpreted. They should also either provide a citation to a more detailed description of the statistic or provide such a description in the Supplementary Information.

Our apologies for not having provided a reference to the cNRI score; this has been added to the methods (Pencina et al. Stat Med 2011, PMID: 21204120). The Methods have also been updated to read: “**The continuous NRI is the sum of two scores: the improvement in classification of a positive event (here, mortality) by the tested model, plus the improvement in classification of a negative event (here, survival) by the tested model. Each improvement has a possible range of [-1,1], so the full cNRI has a possible range of [-2,2]. A score of -2 would mean that every prediction is made worse by the addition of the tested model; a score of 2 means that every prediction is made more accurate by the addition of the tested model.**”

Second, I am grateful to the authors for quantifying the improvement that results from incorporating gene expression into the predictive models. However, I would like to see some additional discussion of the significance of these findings. Do the results suggest that gene expression can produce a big enough increase in predictive accuracy to justify using it in the clinic? Or is the improvement too small? If it is too small to be clinically useful, are there other reasons the findings are important? Perhaps they provide some insight into the underlying etiology of sepsis? The paper could be strengthened by discussing the significance/implications of the results in more detail.

The discussion has been updated to read, “**The impact of the addition of the severity score to clinical practice could be substantial. If envisioned as a rule-out test for mortality (e.g. setting the threshold at a 95% sensitivity), the Duke and Stanford scores showed large increases in specificity (13-21 percentage point absolute increase) compared with standard clinical severity scores alone.**”

In terms of etiology, as we write in the discussion, “The goal of this study was to generate predictive models but not necessarily to define sepsis pathophysiology. However, our community approach identified a large number of genes associated with sepsis mortality that may point to underlying biology. The association with immature neutrophils and inflammation in sepsis has been previously shown¹. Results of this study confirm this finding as we note increases in the neutrophil chemoattractant IL-8 as well as neutrophil-related antimicrobial proteins (*DEFA4*, *BPI*, *CTSG*, *MPO*). These azurophilic granule proteases may indicate the presence of very immature neutrophils (metamyelocytes) in the blood². Many of these genes

have also been noted in the activation of neutrophil extracellular traps (NETs)^{3,4}. NET activation leads to NETosis, a form of neutrophil cell death that can harm the host⁴. Whether these involved genes are themselves harmful or are markers of a broader pathway is unknown.” The addition of the extra GSEA based pathway analysis requested by Reviewer 3 has showed enrichment for additional infectious diseases and inflammation related pathways along with the enrichment for cell cycle and neurogenesis pathways in our pooled genesets.

Reviewer #3 (Remarks to the Author):

On comment 1:

*The authors have now included further methodological details in the supplement as suggested. However, the lack of overlap between models remains a major concern and major limitation to the study (certainly not only to some). No attempt was made to provide objective arguments and interpretations on their study design as well as analysis other than a certain degree of circular logic. Sure, multicollinearity is a notorious problem in the analysis of high-dimensional data that undoubtedly influences parameter estimation due to inflation of variance in regression parameters. It becomes a severe problem in disparate analysis when a model is trained on a first set of selected studies, and evaluated in another set of selected studies of unknown collinearity structure. The motivation to dismiss as “red herring” the concepts in combining blood genomic datasets for proper meta-analysis of gene expression data in sepsis, which would have allowed for better handling of the differing collinearity structures across different datasets, is unclear and quite subjective. If one considers the authors’ stance on the lack of overlap between models, then what is the difference between this study and simply combining previously derived and published gene expression markers for possible prognosis, for example genes and/or gene sets from Wong HR et al. *Physiol Genomics*. 2007;30:146–155 and Dolinay T et al. *Am J Respir Crit Care Med*. 2012;185:1225–1234 and Tsalik EL et al. *Genome Med*. 2014;6:111 and Almansa R et al. *J Infect*. 2015;70:445–456 and Scicluna BP et al. *Am J Respir Crit Care Med*. 2015 Oct 1;192(7):826-35, into one predictive model? Not much. Ultimately, the authors do not provide a potential predictive tool as they sought, but rather showed that by analyzing blood genomic data derived from different studies and analyzed in disparate fashion, using different statistical models and their unaddressed baggage of assumptions on the same discovery/validation sets, revealed substantial discrepancies between models that did not outperform other candidate prognostic biomarkers such as pro-adrenomedullin. The authors should be more clear about these points in their discussion and conclusion.*

The reviewer has raised several points. First, we did not dismiss the idea of meta-analysis; in fact, we performed meta-analysis as a variable selection method in both the Sage and Stanford data pipelines (see Supplemental Method pages 6-8 and Supplemental Figures 2-3). Second, there are several key differences between this paper and prior papers which have examined gene expression in severity/mortality in sepsis, namely: (1) we built prognostic models, instead of just listing genes associated with mortality, unlike most (but not all) papers; (2) we extensively validated our findings in nine external validation cohorts; and (3) instead of using a single analysis methodology, as is traditionally done in biomedical research, we have broadly sampled the possible solution space, suggesting a rough upper bound on performance independent of a specific analytic framework.

The Reviewer suggests that we combine all genes from prior papers that have examined sepsis mortality. Notably, several of the studies the reviewer suggested do not examine sepsis mortality, but compare sepsis to non-infectious inflammation; these have been excluded. The list of references that examine gene expression in sepsis severity or mortality is below. The Supplemental Methods now reads:

“In order to contrast the present findings with prior published results, we searched for all papers that examined transcriptome-wide changes in sepsis associated with increasing severity or mortality. In each of these papers, genes were classified as ‘over-expressed’ or ‘under-expressed’ in association with increasing severity. We took the union of all these differentially expressed genes as inputs, and took the difference of geometric means of these two sets to make a single score. We then measured the AUROC for prediction of mortality using this composite score, and compared to the present scores. To compare this level of performance to the four current models, we used matched t-tests, as well as calculating the mean difference in AUROCs. We did this in the validation cohorts only to prevent bias.”

The Results now read:

“Using the validation and HAI cohorts, we compared the present models to a single prognostic model made with all genes previously associated with mortality (see Supplemental Methods). We found that that 3 of the 4 models show substantial improvement (average increase of roughly 0.1) compared to the prior models; this reached significance ($p < 0.05$) for the Duke and Stanford models (Supplementary Table 3).”

Supplementary Table 3 has been added:

“We found a total of 119 over-expressed and 1,164 under-expressed unique genes previously associated with mortality, which we assessed for prognostic accuracy in the validation datasets. We then compared the results to the output from the four models using paired t-tests.”

Dataset	AUROC of combined 1,273 genes
EMTAB4421	0.581
GSE21802	0.679
GSE 33341	0.969
GSE 54514	0.761
GSE 63990	0.68
Duke HAI	0.83
Glue Burns D1-D30	0.417
Glue Trauma D1-D30	0.958
UF P50 12H	0.624

	Duke	Sage LR	Sage RF	Stanford
mean difference	0.108	0.092	-0.052	0.117
P value	0.046	0.059	0.595	0.014

Perhaps more importantly, the present gene sets could actually be translated into clinical practice. The smallest gene set (Stanford, 12 genes) could be run on existing equipment in a

reasonable turnaround time. There is no comparable platform that could measure 1283 genes in a short timeframe. Thus, in addition to superior prognostic power, the present gene sets are capable of making an actual clinical impact.

Finally, the reviewer brings up pro-adrenomedullin. As we note in the discussion, “The derived discriminatory power of the gene models (AUCs near 0.85) are at least similar to the AUC of proadrenomedullin (0.83) in a recent large prospective trial (TRIAGE study).” Although we do not provide a head-to-head comparison of the given gene sets to pro-ADM, an AUROC of ~0.87-0.89 in validation datasets is well within keeping of this well-studied, but still experimental, biomarker. To our knowledge, no sepsis guideline includes pro-ADM, and it has not been FDA cleared. On the other hand, the results of both pro-ADM and the gene expression biomarkers look quite promising; perhaps eventually they will be additively prognostic. Only further study will tell.

References for papers examining gene expression in sepsis severity and mortality:

- 1 Pachot, A. *et al.* Systemic transcriptional analysis in survivor and non-survivor septic shock patients: a preliminary study. *Immunol Lett* **106**, 63-71, doi:10.1016/j.imlet.2006.04.010 (2006).
- 2 Wong, H. R. *et al.* Genome-level expression profiles in pediatric septic shock indicate a role for altered zinc homeostasis in poor outcome. *Physiol Genomics* **30**, 146-155, doi:10.1152/physiolgenomics.00024.2007 (2007).
- 3 Parnell, G. P. *et al.* Identifying key regulatory genes in the whole blood of septic patients to monitor underlying immune dysfunctions. *Shock* **40**, 166-174, doi:10.1097/SHK.0b013e31829ee604 (2013).
- 4 Almansa, R. *et al.* Transcriptomic correlates of organ failure extent in sepsis. *J Infect* **70**, 445-456, doi:10.1016/j.jinf.2014.12.010 (2015).
- 5 Tsalik, E. L. *et al.* An integrated transcriptome and expressed variant analysis of sepsis survival and death. *Genome Med* **6**, 111, doi:10.1186/s13073-014-0111-5 (2014).
- 6 Bauer, P. R. *et al.* Diagnostic accuracy and clinical relevance of an inflammatory biomarker panel for sepsis in adult critically ill patients. *Diagn Microbiol Infect Dis* **84**, 175-180, doi:10.1016/j.diagmicrobio.2015.10.003 (2016).

On comment 2:

The authors have addressed the comment

On comment 3:

The authors have addressed the comment

On comment 4:

The authors have not addressed a question on their pathway analysis methods. Why haven't the authors considered using highly curated databases for pathway analysis such as GSEA or Ingenuity? The authors should at the least provide pathway enrichment results based on these databases.

As per reviewer's suggestion we expanded our analyses to include KEGG along with GO and Reactome. Also, to overcome some of the pitfalls of gene-based over-representation analysis, we performed an expression-based enrichment analysis (GSEA) using the pooled 58 gene sets from

all four models. The main text and the supplementary Table 12 were modified to reflect these changes. The Results were heavily modified, and the section now reads:

“We next tested the 58 genes for enrichment in curated gene sets from gene ontologies, Reactome and KEGG pathways using two different enrichment methodologies: gene-based over-representation analysis and expression-based GSEA. After multiple hypothesis testing corrections, 4 out of 3330 gene sets tested were significantly over-represented at an FDR of 5% (Supplementary Table 12a). These include genes related to the regulation of T cell activation and proliferation, cytokine-mediated signaling pathway and RHO GTPases activation of CIT. The relatively low number of pathways enriched in over-representation analysis may be due to the low number of genes in the predictor set. Enrichment of 58 gene predictors’ expression were also tested using GSEA. 546 out of 1576 curated pathways were enriched at an FDR of 5%; top pathways are shown in Supplementary Table 12b. A brief examination of enriched pathways activated in non-survivors showed mostly inflammation-related pathways, while survivors showed largely developmental pathways. Since the models were generated in a way that penalized the inclusion of genes that were redundant for classification purposes, and since genes redundant for classification purposes are often from the same biological pathway, their exclusion from the models limits the utility of enrichment analyses.”

Due to the proprietary nature of IPA, this method was not utilized.

REVIEWERS' COMMENTS:

Reviewer #3 (Remarks to the Author):

The authors did a good job in responding to my comments. I have no further suggestions that might improve the manuscript.

Reviewer #3 (Remarks to the Author):

The authors did a good job in responding to my comments. I have no further suggestions that might improve the manuscript.

No response required. We are pleased with the review process.